

# Association of FKBP5 polymorphisms with patient susceptibility to coronary artery disease comorbid with depression

Haidong Wang[1,*], Chao Wang[2,*], Xingfa Song[1], Hai Liu[1], Yun Zhang[1] and Pei Jiang[3]

[1] Department of Pharmacy, The Affiliated Lianyungang Hospital of Xuzhou Medical University/The First People's Hospital of Lianyungang, Lianyungang, Jiangsu, China
[2] Department of Pharmacy, Hainan General Hospital, Hainan Affiliated Hospital of Hainan Medical University, Haikou, Hainan, China
[3] Department of Pharmacy, Affiliated Jining First People's Hospital of Jining Medical University, Jining Medical University, Jining, Shandong, China
[*] These authors contributed equally to this work.

## ABSTRACT

**Background**. Coronary artery disease (CAD) and depression cause great burden to society and frequently co-occur. The exact mechanisms of this comorbidity are unclear. FK506-binding protein 51 (FKBP51) is correlated with cardiovascular disease and depression. The aim of this study was to determine the role of the seven single nucleotide polymorphisms (SNPs) of *FKBP5* that code FKBP51, namely, rs1360780 (C>T), rs2817032 (T>C), rs2817035 (G>A), rs9296158 (G>A), rs9470079 (G>A), rs4713902 (T>C), and rs3800373 (C>T) in a patient's susceptibility to comorbid CAD and depression.

**Methods**. We enrolled 271 Northern Chinese Han patients with CAD, including 123 patients with depression and 147 patients without depression. We also included 113 healthy controls that match the patients' sex and age. Genomic DNA from whole blood was extracted, and seven SNPs were assessed using MassArray method. Patient Health Questionnaire-9 was applied to access the depression.

**Results**. The GA genotype for rs9470079 was associated with a significantly decreased risk of CAD (odds ratio = 0.506, 95% confidence interval = 0.316–0.810, $P = 0.005$) when the GG genotype was used as reference. A statistically significant difference was observed among females but not among males in the rs9470079 genotype and allele frequency. Patients with CAD were further divided into CAD+D and CAD-D groups according to the presence of comorbid depression and were compared with the controls. Significant differences were found regarding the genotype and allele frequency of rs2817035 and rs9470079 in CAD+H groups compared with the control subjects in all groups and the female groups ($P < 0.05$).

**Conclusions**. The current study found a remarkable association between *FKBP5* gene variations and the risk of comorbid CAD and depression in a north Chinese population. rs9470079 may be a potential gene locus for the incidence of comorbid CAD and depression.

Corresponding author
Pei Jiang, jiangpeicsu@sina.com

## INTRODUCTION

Coronary artery disease (CAD) is a major public health challenge globally; CAD is responsible for approximately 32% of deaths worldwide, which exceeds that of all cancers combined in most developed countries (*GBD 2016 Causes of Death Collaborators, 2017*; *Benjamin et al., 2017*) Depression is a psychological complication that may occur alongside CAD; it is an under-recognized determinant of outcomes in patients with CAD because of its high sudden death rate and poor prognostic association with CAD (*Huffman et al., 2013*; *Raison, Capuron & Miller, 2006*). Major depressive disorder and minor depression affect 20% and 30%–45% of patients with CAD, respectively (*Baghai et al., 2018*).

The direction and cause mechanism of the association between CAD and depression remain unclear. However, many studies have demonstrated their shared risk factors, including hypercortisolemia, inflammation, autonomic arousal, serotonin signaling-altered platelet function, and hypothalamus–pituitary–adrenocortical (HPA) axis dysfunction (*Lett et al., 2004*). Genetic factors contribute to the comorbidity of CAD and depression. Genome-wide association study (GWAS), a non-hypothesis-driven and unbiased approach, is a standard tool used to analyze the potential associations between the traits of a disease and single nucleotide polymorphisms (SNPs). Numerous GWASs have been implemented to investigate CAD and depression and involved tens of thousands of case and controls from a great range of geographic, demographic, and ethnic backgrounds. (*Guo et al., 2017*; *Nurnberg et al., 2016*; *Ormel, Hartman & Snieder, 2019*) Over 60 CAD loci were identified for CAD susceptibility (*Nikpay et al., 2015*). According to a GWAS of depression in 2018, 17 variants in excitatory synaptic pathways were identified by a UK Biobank study (*Howard et al., 2018*). However, no GWAS of comorbid CAD and depression has been reported. By contrast to GWAS, candidate gene study, which is an approach based on hypothesis, has been applied to uncover the genetic basis of susceptibility to diseases. For example, some gene studies have revealed the association of comorbid CAD and depression with genetic defects in plasminogen activator inhibitor 1, 5-hydroxytryptamine, and apolipoprotein E (*Fritze et al., 2011*; *Golimbet et al., 2012*; *Lahlou-Laforet et al., 2006*).

FK506-binding protein (FKBP) is coded by the *FKBP* gene, which is located on chromosome 6. FKBP51 is an important member of the FKBP protein family and is coded by *FKBP5*. FKBP5 is a vital modulator that regulates the amount of biological processes in the periphery and the brain and is a regulator of glucocorticoid receptors (GRs), which are associated with the HPA axis function (*Appelhof et al., 2006*). Glucocorticoids can increase *FKBP5* gene expression in various tissues in a dose-dependent manner (*Lee et al., 2018*). GR condition and HPA axis function are closely related to the pathogenesis of CAD and depression (*Dickens, 2015*). Systematic reviews and meta-analysis studies have proven that the SNPs of *FKBP5* are associated with depression (*Normann & Buttenschon, 2019*; *Piechaczek et al., 2019*). FKBP5 expression is associated with insulin resistance, type 2 diabetes, and obesity, which are closely related to cardiovascular disease (*Fichna et al., 2018*; *Sidibeh et al., 2018*). The regulation of FKBP5 may be associated with cardiometabolic risk (*Ortiz et al., 2018*; *Zannas et al., 2019*). Thus, we can reasonably assume that *FKBP5* is associated with CAD susceptibility.

Given its associations with CAD and depression, we speculated that the *FKBP5* gene may be the gene underlying the comorbidity of CAD and depression. This study aimed to investigate the association of *FKBP5* polymorphisms with the susceptibility to comorbid depression in patients with CAD from a Northern Chinese population. Seven SNPs, namely, rs1360780 (C>T), rs2817032 (T>C), rs2817035 (G>A), rs9296158 (G>A), rs9470079 (G>A), rs4713902 (T>C), and rs3800373 (C>T), were selected. Their correlation with the comorbidity was evaluated.

## METHODS

### Subjects

This study recruited participants from the First Peoples' Hospital of Jining between February 2016 and May 2018. A total of 270 northern Han Chinese patients with CAD and 113 healthy controls matched with patients' sex and age were enrolled in this study. All patients with CAD were diagnosed by experienced cardiologists on the basis of significant standards: angiographic evidence of luminal diameter narrowing >50% in at least one main coronary artery, previous history of coronary artery bypass graft surgery, and history of percutaneous coronary intervention. Patients with renal failure, congenital heart disease, tumors, immune system disorders, malignancies, congenital heart disease, and infectious heart disease were excluded. CAD patients with or without depression were assessed by at least two experienced psychiatrists on the basis of the fifth edition of the Diagnostic and Statistical Manual of Mental Disorders for depressive disorder, characterized by significant anhedonia and depressed mood. Patient Health Questionnaire-9 (PHQ-9), a commonly used 9-item questionnaire, was used to assess the severity of depressive symptoms. A score that was equal or greater than 5 was used as the cutoff score for depression (*Duko et al., 2018*).

Health controls were selected from the physical examination program through clinical examination and electrocardiogram at the same period. This study was designed in accordance with the Declaration of Helsinki and approved by the ethics committee of the First Peoples' Hospital of Jining (approval number: JY2016035). All subjects provided written informed consent.

### DNA isolation and genotyping

About 1 ml of peripheral blood was collected and extracted from the subjects using a TIANamp Blood DNA Kit (TIANGEN, China) according to the manufacturer's instructions. The concentration and purity of DNA samples were detected with NanoDrop-1000 (NanoDrop, USA) to ensure that the samples were available for subsequent experiments. All DNA samples were genotyped through polymerase chain reaction (PCR)–ligase detection reaction. PCR of the four target single-nucleotide polymorphisms was amplified by the primers listed in Table 1 from each participant. The samples were processed by shrimp alkaline phosphatase, extended, and purified using iPLEX extension reagents (Agena Bioscience, USA) and Nanodispenser RS1000. Matrix-assisted laser desorption/ionization time-of-flight mass spectrometry was conducted to detect the primer extension products, and Spectro-Typer was used to automatically analyze the

**Table 1   Primers of FKBP5 genes used in the PCR.**

| SNP | Ancestor allele | Primer sequence | Product size |
|---|---|---|---|
| rs1360780 | C | 5′-ACGTTGGATGTGCCAGCAGTAGCAAGTAAG-3′<br>5′-ACGTTGGATGCAGGCACAGAAGGCTTTCAC-3′ | 88 |
| rs2817032 | T | 5′- ACGTTGGATGTTTCACAGGTACCCCATTCC-3′<br>5′-ACGTTGGATGAATATCACAGGCTTGCTGGG-3′ | 103 |
| rs2817035 | G | 5′-ACGTTGGATGGTTGCAAACAGAGGTAGGAG-3′<br>5′- ACGTTGGATGCTCTTTTCTCCTAGGATCCC-3′ | 99 |
| rs9296158 | G | 5′- ACGTTGGATGGACCTGGTAATATCACTCTC-3′<br>5′- ACGTTGGATGCTGGGCTAGGGGTAATTCAA-3′ | 118 |
| rs9470079 | G | 5′- ACGTTGGATGGCCTCCCAAAATGCTATATC-3′<br>5′-ACGTTGGATGATACCATACTCTAGGCTGGG-3′ | 104 |
| rs4713902 | T | 5′-ACGTTGGATGGGAGCCAAAACATGAAGAGC-3′<br>5′-ACGTTGGATGTAGGCAACCTGTATAAGCTG-3′ | 99 |
| rs3800373 | C | 5′-ACGTTGGATGTGACTTTTTAGTACTAAGC-3′<br>5′-ACGTTGGATGCCCTAGTGTAGAAGAGCAAC-3′ | 101 |

genotyping data. More than 10% of the samples were randomly selected and retested to verify the validity of MassARRAY results.

## Statistical analysis

All genotyping results of the investigated patients and controls were tested for Hardy–Weinberg equilibrium (HWE) by applying the chi-square test ($\chi 2$ test). Differences in genotypic distributions and allele frequencies in the cases and controls were compared among groups for statistical significance through chi-square statistics ($\chi 2$ test). The associations between the genotypes and CAD/CAD with comorbid depression were evaluated via the odds ratio (OR), with 95% confidence interval (CI). A two-sided $p$ value below 0.05 was considered statistically significant. All statistical analyses were performed with SPSS 17.0 for Windows (SPSS Inc., Chicago, IL, USA).

## RESULTS

Table 2 shows the demographic and clinical characteristics of the participants in this study., No significant differences were found between the CAD and the health control groups in term of age, gender, smoking, drinking and body mass index (BMI) ($P > 0.05$). No significant differences were observed when the CAD cases were subdivided to the CAD with depression (CAD+D) and CAD without depression (CAD-D) groups based on comorbid depression.

The results demonstrated that the seven observed genotype frequencies were in accordance with the HWE ($P \geq 0.088$). The genotypic distribution and allele frequencies of the seven genetic polymorphisms between the CAD and control groups in all participants, male participants only, and female participants only were compared (Table 3 for all participants, Table 4 for female participants with significant results and Table S1 for all male participants and all female participants). No statistically significant difference was observed between the patients with CAD and controls for the genotypic and allelic distributions

**Table 2   Demographic and clinical characteristics of the participants.**

| Variables | CAD (n = 270) | Controls (n = 113) | P-value[a] | CAD+D (n = 123) | P-value[b] | CAD-D (n = 147) | P-value[c] |
|---|---|---|---|---|---|---|---|
| Age (yrs) | 56.2 ± 10.4 | 52.9 ± 10.2 | 0.887 | 57.1 ± 10.2 | 0.977 | 55.5 ± 10.5 | 0.820 |
| Gender (M/F, n) | 128/142 | 52/61 | 0.804 | 53/70 | 0.651 | 75/72 | 0.424 |
| Smoking (n, %) | 88, 32.5 | 28, 24.8 | 0.129 | 36, 29.3 | 0.438 | 52, 35.3 | 0.067 |
| Drinking (n, %) | 93, 34.4 | 33, 29.2 | 0.319 | 35, 28.4 | 0.899 | 58, 39.4 | 0.086 |
| BMI (kg/m²) | 24.3 ± 3.3 | 24.0 ± 3.0 | 0.205 | 24.4 ± 3.6 | 0.119 | 24.2 ± 2.9 | 0.983 |

Notes.

[a] CAD versus Controls.

[b] CAD+D versus Controls.

[c] CAD+D versus CAD-D.

CAD, coronary heart disease; CAD+D, CHD with depression; CAD-D, CAD without depression.

of the rs1360780 (C>T), rs2817032 (T>C), rs2817035(G>A), rs9296158(G>A) and rs3800373(C>T) polymorphisms in the investigated group, including all, male or female participants.

The TC+CC genotype frequency for rs4713902 was significantly higher in the CAD cases than in the controls ($P = 0.049$). For rs9470079, the GA and GA+AA genotypes were associated with significantly decreased risks of CAD (OR = 0.506, 95% CI [0.316–0.810], $p = 0.005$ and OR = 0.502, 95% CI [0.320–0.788], $p = 0.003$ for GA and GA+AA, respectively) when the GG genotype was used as the reference The A allele showed a significant association with the CAD group (OR = 0.626, 95% CI [0.450–0.871], $p = 0.005$).

Interestingly, no statistically significant difference was found in males in terms of the rs9470079 genotype and allele frequency, whereas a statistically significant difference was observed among females (OR = 0.419, 95% CI [0.220–0.799], $p = 0.008$ for GA vs. GG; OR = 0.406, 95% CI [0.219–0.752], $p = 0.004$ for GA+AA vs. GG; OR = 0.545, 95% CI [0.344–0.862], $p = 0.010$ for A vs. G).

The patients with CAD were divided into the CAD+D and CAD-D groups based on comorbid depression, and all the investigated genotype and allele frequency distributions of polymorphisms were compared within the CAD+D, CAD-D, and healthy groups. The genotype frequencies of the subgroups were compared with those of the controls, and the results are presented in Table 5, No significant associations were observed in rs1360780 (C>T), rs2817032 (T>C), rs2817035(G>A) and rs3800373(C>T) SNPs and CAD of the subgroups ($P > 0.05$). A significant difference in the genotype and allele frequencies of rs2817035 and rs9470079 was noted in the CAD+H groups compared with the control subjects ($P < 0.05$ for both comparisons). A significant difference was found in the allele frequency of the rs4713902 polymorphism in the CAD+H groups compared with the control subjects ($P = 0.034$).

Stratification comparison by gender was performed for genotype and allele frequencies of rs9470079 in the CAD+D, CAD-D and healthy groups. The results present in Table 6. The combination of the rs9470079 polymorphism was not associated with CAD comorbid depression or not in male group. However, significant differences were observed in the genotype frequency of rs9470079 in both CAD+D and CAD-D groups compared with

**Table 3  Genotypic and allelic distribution of seven FKBP5 genes between all CAD patients ($n = 270$) and Controls ($n = 113$).**

| SNP | Genotype/ allele | Case,(%) | Control,(%) | P value[a] ($\chi^2$) | OR (95% CI) | P value[b] |
|---|---|---|---|---|---|---|
| rs1360780 | CC | 134 (49.6) | 60 (53.1) | 0.644 (0.879) | 1.00 | Referent |
| | CT | 121 (44.8) | 49 (43.4) | | 1.106 (0.705–1.735) | 0. 622 |
| | TT | 15 (5.6) | 4 (3.5) | | 1.679 (0.535–5.272) | 0.375 |
| | CT+TT | 136 (50.4) | 53 (46.9) | 0.536 (0.383) | 1.149 (0.740–1.784) | 0.563 |
| | C | 389 (72.0) | 169 (74.8) | 0.437 (0.605) | 1.00 | Referent |
| | T | 151 (28.0) | 57 (25.2) | | 1.151 (0.808–1.640) | 0.151 |
| rs2817032 | TT | 142 (52.6) | 69 (61.1) | 0.253 (2.748) | 1.00 | Referent |
| | TC | 108 (40.0) | 39 (34.5) | | 1.346 (0.845–2.144) | 0.211 |
| | CC | 20 (7.4) | 5 (4.4) | | 1.944 (0.700–5.397) | 0.202 |
| | TC+CC | 128 (47.4) | 44 (38.9) | 0.129 (2.310) | 1.414 (0.904–2.211) | 0.129 |
| | T | 392 (72.6) | 177 (78.3) | 0.098 (2.734) | 1.00 | Referent |
| | C | 148 (27.4) | 49 (21.7) | | 1.364 (0.943–1.972) | 0.099 |
| rs2817035 | GG | 121 (44.8) | 61 (54.0) | 0.040 (6.429)[*] | 1.00 | Referent |
| | GA | 138 (51.1) | 52 (46.0) | | 1.338 (0.859–2.084) | 0.198 |
| | AA | 11 (4.1) | 0 (0.0) | | – | – |
| | GA+AA | 149 (55.2) | 52 (46.0) | 0.101 (2.685) | 1.445 (0.930–2.245) | 0.102 |
| | G | 380 (70.4) | 174 (77.0) | 0.062 (3.489) | 1.00 | Referent |
| | A | 160 (29.6) | 52 (23.0) | | 1.409 (0.982–2.021) | 0.062 |
| rs9296158 | GG | 112 (41.5) | 47 (41.6) | 0.994 (0.013) | 1.00 | Referent |
| | GA | 121 (44.8) | 51 (45.1) | | 0.996 (0.621–1.597) | 0.985 |
| | AA | 37 (13.7) | 15 (13.3) | | 1.035 (0.519–2.064) | 0.922 |
| | GA+AA | 158 (58.5) | 66 (58.4) | 0.984 (0.000) | 1.005 (0.643–1.569) | 0.984 |
| | G | 345 (63.9) | 145 (64.2) | 0.943 (0.005) | 1.00 | Referent |
| | A | 195 (36.1) | 81 (35.8) | | 1.012 (0.732–1.398) | 0.943 |
| rs9470079 | GG | 146 (54.1) | 42 (37.2) | 0.010 (9.121)[*] | 1.00 | Referent |
| | GA | 102 (37.8) | 58 (51.3) | | 0.506 (0.316–0.810) | 0.005[*] |
| | AA | 22 (8.1) | 13 (11.5) | | 0.487 (0.226–1.048) | 0.066 |
| | GA+AA | 124 (45.9) | 71 (62.8) | 0.003 (9.110)[*] | 0.502 (0.320–0.788) | 0.003[*] |
| | G | 394 (73.0) | 142 (62.8) | 0.005 (7.783)[*] | 1.00 | Referent |
| | A | 146 (27.0) | 84 (37.2) | | 0.626 (0.450–0.871) | 0.005[*] |
| rs4713902 | TT | 145 (53.7) | 73 (64.6) | 0.114 (4.351) | 1.00 | Referent |
| | TC | 109 (40.4) | 33 (29.2) | | 1.663 (1.029–2.688) | 0.038 |
| | CC | 16 (5.9) | 7 (6.2) | | 1.151 (0.435–2.921) | 0.768 |
| | TC+CC | 125 (46.3) | 40 (35.4) | 0.049 (3.858)[*] | 1.573 (0.999–2.477) | 0.050[*] |
| | T | 399 (73.9) | 179 (79.2) | 0.119 (2.430) | 1.00 | Referent |
| | C | 141 (26.1) | 47 (20.8) | | 1.346 (0.999–2.477) | 0.120 |

**Table 3** (*continued*)

| SNP | Genotype/ allele | Case,(%) | Control,(%) | P value[a] ($\chi^2$) | OR (95% CI) | P value[b] |
|-----|------------------|----------|-------------|----------------|-------------|-----------|
| rs3800373 | CC | 72 (26.7) | 36 (31.9) | 0.422 (1.724) | 1.00 | Referent |
| | CA | 182 (67.4) | 73 (64.6) | | 2.000 (0.623–6.421) | 0.244 |
| | AA | 16 (5.9) | 4 (3.5) | | 1.247 (0.769–2.022) | 0.372 |
| | CA+AA | 198 (73.3) | 77 (68.1) | 0.980 (0.001) | 0.994 (0.607–1.625) | 0.980 |
| | C | 326 (60.4) | 145 (64.2) | 0.326 (0.966) | 1.00 | Referent |
| | A | 214 (39.6) | 81 (35.8) | | 1.175 (0.852–1.621) | 0.326 |

**Notes.**

Abbreviations: CI, confidence interval; OR, odds ratio.

*$P < 0.05$.

[a]$P$ value for genotype and allelefrequencies in cases and controls using 2-sided $\chi^2$ test.

[b]$P$ values adjusted by age and genderusing logistic regression.

**Table 4** Genotypic and allelic distribution of FKBP5 (rs9470079) genes between female CAD patients ($n = 142$) and Controls ($n = 61$).

| SNP | Genotype/ allele | Case,(%) | Control,(%) | P value[a] ($\chi^2$) | OR (95% CI) | P value[b] |
|-----|------------------|----------|-------------|----------------|-------------|-----------|
| rs9470079 | GG | 85 (59.9) | 23 (37.7) | 0.014 (8.540)* | 1.00 | Referent |
| | GA | 58 (40.8) | 31 (50.8) | | 0.419 (0.220–0.799) | 0.008* |
| | AA | 9 (6.3) | 7 (11.5) | | 0.348 (0.117–1.035) | 0.058 |
| | GA+AA | 57 (47.1) | 38 (62.3) | 0.004 (8.412)* | 0.406 (0.219–0.752) | 0.004* |
| | G | 210 (73.9) | 77 (63.1) | 0.009 (6.804)* | 1.00 | Referent |
| | A | 66 (26.1) | 45 (36.9) | | 0.545 (0.344–0.862) | 0.010* |

**Notes.**

Abbreviations: CI, confidence interval; OR, odds ratio.

*$P < 0.05$.

[a]$P$ value for genotype and allelefrequencies in cases and controls using 2-sided $\chi^2$ test.

[b]$P$ values adjusted by age and genderusing logistic regression.

the female control group ($P = 0.039$ and $P = 0.036$, respectively). Allele frequency of the rs9470079 polymorphism was significantly different in the CAD+D and CAD-D groups compared with the female control group ($P = 0.019$ and $P = 0.013$, respectively).

## DISCUSSION

FKBP51 is a FK506-binding protein with high molecular weight and is coded by the *FKBP5* gene, which consists of 13 exons located on chromosome 6 (6p21.31). FKBP51 has important roles in the pathogenesis of psychological complications, such as depression, obsessive–compulsive disorder, and schizophrenia (*Daskalakis & Binder, 2015*; *Ferrer et al., 2018*). FKBP51 affects GR activity by reducing its binding affinity and regulating the HPA axis. FKBP51 can inhibit other steroid hormone receptors, including progesterone and androgen receptors (*Jaaskelainen, Makkonen & Palvimo, 2011*). The conditions of GRs, HPA axis, and steroid hormone receptors are related to the pathogenesis of CAD. Some studies reported an association between the *FKBP5* gene and cardiovascular risk.

GWAS is a powerful way to identify the genes involved in human disease, but this approach has not detected the effects of the FKBP5 locus (*Hähle et al., 2019*). However, FKBP5 gene variations have been associated with risks for varying disorders. Thus, we investigated the association of *FKBP5* gene polymorphisms with the susceptibility of patients with CAD in a northern Chinese population. The GA and GA+AA genotypes of

**Table 5  Genotypic and allele Distribution of seven FKBP polymorphisms among the CAD with depression group, CAD without depression group and control group.**

|  | SNP | 1 | 2 | 3 | P-value | | |
|---|---|---|---|---|---|---|---|
|  |  | CAD $^{+}$H (n = 123) | CAD$^{-}$H (n = 147) | Control (n = 113) | 1vs.2 | 1vs.3 | 2vs.3 |
| rs1360780 | CC | 60 (48.8) | 74 (50.3) | 60 (53.1) | 0.968 | 0.650 | 0.739 |
|  | CT | 56 (45.5) | 65 (44.2) | 49 (43.4) |  |  |  |
|  | TT | 7 (5.7) | 8 (5.5) | 4 (3.5) |  |  |  |
|  | C | 176 (48.8) | 213 (48.8) | 169 (48.8) | 0.816 | 0.429 | 0.551 |
|  | T | 70 (48.8) | 81 (48.8) | 57 (48.8) |  |  |  |
| rs2817032 | TT | 63 (51.2) | 79 (53.7) | 69 (61.1) | 0.733 | 0.301 | 0.334 |
|  | TC | 52 (42.3) | 56 (38.1) | 39 (34.5) |  |  |  |
|  | CC | 8 (6.5) | 12 (8.2) | 5 (4.4) |  |  |  |
|  | T | 178 (48.8) | 214 (48.8) | 177 (48.8) | 0.991 | 0.134 | 0.148 |
|  | C | 68 (48.8) | 80 (48.8) | 49 (48.8) |  |  |  |
| rs2817035 | GG | 52 (42.3) | 67 (45.6) | 61 (54.0) | 0.665 | 0.021* | 0.098 |
|  | GA | 65 (52.8) | 75 (51.0) | 52 (46.0) |  |  |  |
|  | AA | 6 (4.9) | 5 (3.4) | 0 (0.0) |  |  |  |
|  | G | 169 (48.8) | 209 (48.8) | 174 (48.8) | 0.546 | 0.043* | 0.130 |
|  | A | 77 (48.8) | 85 (48.8) | 52 (48.8) |  |  |  |
| rs9296158 | GG | 53 (43.1) | 59 (40.1) | 47 (41.6) | 0.590 | 0.903 | 0.865 |
|  | GA | 56 (45.5) | 65 (44.2) | 51 (45.1) |  |  |  |
|  | AA | 14 (11.4) | 23 (15.7) | 15 (13.3) |  |  |  |
|  | G | 162 (48.8) | 183 (48.8) | 145 (48.8) | 0.385 | 0.700 | 0.654 |
|  | A | 84 (48.8) | 111 (48.8) | 81 (48.8) |  |  |  |
| rs9470079 | GG | 71 (57.7) | 75 (51.0) | 42 (37.2) | 0.462 | 0.006* | 0.083 |
|  | GA | 44 (35.8) | 58 (39.5) | 58 (51.3) |  |  |  |
|  | AA | 8 (6.5) | 14 (9.5) | 13 (11.5) |  |  |  |
|  | G | 186 (48.8) | 208 (48.8) | 142 (48.8) | 0.205 | 0.003* | 0.056 |
|  | A | 60 (48.8) | 86 (48.8) | 84 (48.8) |  |  |  |
| rs4713902 | TT | 62 (50.4) | 83 (56.5) | 73 (64.6) | 0.140 | 0.088 | 0.139 |
|  | TC | 50 (40.7) | 59 (40.1) | 33 (29.2) |  |  |  |
|  | CC | 11 (8.9) | 5 (3.4) | 7 (6.2) |  |  |  |
|  | T | 174 (48.8) | 225 (48.8) | 179 (48.8) | 0.127 | 0.034* | 0.468 |
|  | C | 72 (48.8) | 69 (48.8) | 47 (48.8) |  |  |  |
| rs3800373 | CC | 35 (28.5) | 37 (25.2) | 36 (31.9) | 0.702 | 0.773 | 0.302 |
|  | CA | 82 (66.7) | 100 (68.0) | 73 (64.6) |  |  |  |
|  | AA | 6 (4.8) | 10 (6.8) | 4 (3.5) |  |  |  |
|  | C | 152 (48.8) | 174 (48.8) | 145 (48.8) | 0.538 | 0.594 | 0.248 |
|  | A | 94 (48.8) | 120 (48.8) | 81 (48.8) |  |  |  |

**Notes.**

Abbreviations: CI, confidence interval; OR, odds ratio; CAD+D, CAD with depression; CAD-D, CAD without depression.

[a] P value for genotype and allele frequencies in cases and controls using 2-sided $\chi^2$ test.

[b] P values adjusted by age and gender using logistic regression.

*$P < 0.05$.

**Table 6  Genotypic and Allelic Distribution of FKBP5 (rs9470079) polymorphisms among the three studied groups between different genders.**

| | SNP | 1 | 2 | 3 | P-value | | |
|---|---|---|---|---|---|---|---|
| | | CAD $^+$H | CAD$^-$H | Control | 1vs.2 | 1vs.3 | 2vs.3 |
| Males | | $n = 53$ | $n = 75$ | $n = 52$ | | | |
| | GG | 29 (54.7) | 32 (42.7) | 19 (36.5) | 0.236 | 0.148 | 0.679 |
| | GA | 21 (39.6) | 33 (44.0) | 27 (52.0) | | | |
| | AA | 3 (5.7) | 10 (13.3) | 6 (11.5) | | | |
| | G | 79 (74.5) | 97 (64.7) | 65 (62.5) | 0.094 | 0.060 | 0.724 |
| | A | 27 (25.5) | 53 (35.3) | 39 (37.5) | | | |
| Females | | $n = 70$ | $n = 72$ | $n = 61$ | | | |
| | GG | 42 (60.0) | 43 (59.7) | 23 (37.7) | 0.975 | 0.039[*] | 0.036[*] |
| | GA | 23 (32.9) | 25 (34.7) | 31 (50.8) | | | |
| | AA | 5 (7.1) | 4 (5.6) | 7 (11.5) | | | |
| | G | 107 (76.4) | 111 (77.1) | 77 (63.1) | 0.896 | 0.019[*] | 0.013[*] |
| | A | 33 (23.6) | 33 (22.9) | 45 (36.9) | | | |

**Notes.**
CAD+D, CAD with depression; CAD-D, CAD without depression.
$^*P < 0.05$.

rs9470079 were associated with a remarkably decreased risk of CAD. The exact mechanism underlying the effect of FKBP5 on CAD is unclear, but some reports have provided evidence of the processes involved. The epigenetic upregulation of FKBP5 caused by aging and stress is driven by FKBP5–nuclear factor kappa-light-chain-enhancer of activated B cell signaling, mediates inflammation, and contributes to cardiovascular risk (*Zannas et al., 2019*). *Ortiz et al. (2018)* stated that cardiometabolic risk may be associated with increased DNA methylation of *FKBP5*, which is associated with the risk factors for CAD, such as the higher levels of glycosylated hemoglobin, low-density lipoprotein cholesterol, body mass index, and waist circumference. Moreover, FKBP5 increases platelet expression in patients with myocardial infarction, which mostly occurs because of CAD (*Eicher et al., 2016*).

We further classified the CAD group into CAD+D and CAD-D groups depending on the presence of comorbid depression to investigate the association of *FKBP5* gene polymorphism with susceptibility to CAD with comorbid depression. The genotypes and alleles of rs2817035 and rs9470079 and the alleles of rs4713902 showed significant differences only between the CAD+D and control groups but not between the CAD-D and control groups and between CAD+D and CAD-D groups. Rs4713902 polymorphisms interact with chronic low family support in association with a child's mental health status (*Adrian et al., 2015*). *Ferrer et al. (2018)* reported that individuals with rs9470079—A show a reduced dexamethasone suppression test ratio and suggested a probable effect between the FKBP5 rs9470079 polymorphism and impaired HPA axis negative feedback in major depression. However, we did not find any remarkable differences in these FKBP5 polymorphisms for the CAD+D group compared with the CAD-D and healthy control groups. Other studies focused on the effect of FKBP5 SNPs rs1360780 and rs3800373 on depression. *Normann & Buttenschon (2019)* revealed that rs1360780 possibly moderates the effects of systemic lupus erythematous in depression and that rs3800373 is associated with a

remarkable increased risk of depressive disorders. We failed to demonstrate the association between the CAD+D and CAD-D group or healthy control groups for rs1360780 or rs3800373. These results suggested that common depression and depression comorbid with CAD may have different pathogenetic mechanisms.

CAD is a sex-dependent disease that is two to five times more common in middle-aged men than in their women counterparts; its incidence has decreased in men but has increased in women (*Shively, Musselman & Willard, 2009*; *Yang et al., 2010*). Our results presented a remarkable association between rs9470079 and CAD in the female groups but not in the male groups. Depressive disorders are twice as likely to occur in women than in men (*Gorman, 2006*). Thus, we investigated the genotypic and allelic distributions of the rs9470079 polymorphism in the CAD+D, CAD-D, and control groups between different genders. The results showed a significant difference in the genotypic and allelic distributions of the rs9470079 polymorphisms in the CAD+D and CAD-D groups compared with the controls in the female groups but not in the male groups. Thus, genetics may play different roles in different genders. The present results were consistent with those of a previous study on depression and CAD in Swedish twins (*Kendler et al., 2009*), which demonstrated that genetic sources play a large role in CAD+D comorbidity in women, whereas environmental effects play a large role in CAD-D in men. The *FKBP5* gene contains hormone response elements that can bind receptors to sex hormones (*Magee et al., 2006*). These elements have different levels in males and females and may play a role in the association of CAD with comorbid depression in different genders.

Several limitations of this study had to be mentioned. First, this study only evaluated a small population in northern China, and the sample size was limited. Genetic polymorphisms of ethnic differences may determine varying functions in different populations. Thus, large sample sizes from different groups are required to obtain reliable outcomes. Second, this study only tested seven of the genotypes of *FKBP5* and the tagging of the *FKBP5* gene was incomplete. Thus, this study could not fully reflect the association of the polymorphisms of FKBP5 with comorbid CAD and depression. A previous study spanning the whole *FKBP5* gene showed 18 SNPs in strong linkage disequilibrium among Caucasians (*Zannas et al., 2016*). However, we did not found linkage disequilibrium in our study (data no shown), probably owing to the limited number of samples or the incomplete gene locus. Third, the data on FKPB51 level were insufficient, and we failed to assess the influence of FKBP5 expression on the incidence of comorbid CAD and depression by regulating the FKPB51 level.

## CONCLUSION

The current study proposed a remarkable association between *FKBP5* gene variations and the risk of comorbid CAD and depression in a Northern Chinese population. Rs9470079 may be a potential gene locus for the incidence of comorbid CAD and depression. The present findings should be verified through replication studies on large ethnically disparate specimens and with variants covering the whole gene. The exact role of *FKBP5* gene polymorphisms in the pathogenesis of comorbid CAD and depression requires further investigation.

### Funding

This study was supported by the Taishan Scholar Program of Shandong Province (No. tsqn201812159), the Foundation of Clinical Pharmacy of Chinese Medical Association (No. LCYX-M008) and Excellent Youth Foundation Program for Developing Health by Relying on Science and Education of Lianyungang (No. QN1601). The funders had no role in study design, data collection and analysis, decision to publish, or preparation of the manuscript.

### Grant Disclosures

The following grant information was disclosed by the authors:
Taishan Scholar Program of Shandong Province: tsqn201812159.
the Foundation of Clinical Pharmacy of Chinese Medical Association: LCYX-M008.
Excellent Youth Foundation Program for Developing Health by Relying on Science and Education of Lianyungang:  QN1601.

### Competing Interests

The authors declare there are no competing interests.

### Author Contributions

- Haidong Wang conceived and designed the experiments, prepared figures and/or tables, and approved the final draft.
- Chao Wang and Yun Zhang analyzed the data, prepared figures and/or tables, and approved the final draft.
- Xingfa Song and Hai Liu performed the experiments, authored or reviewed drafts of the paper, and approved the final draft.
- Pei Jiang conceived and designed the experiments, authored or reviewed drafts of the paper, and approved the final draft.

### Human Ethics

The following information was supplied relating to ethical approvals (i.e., approving body and any reference numbers):

The ethics committee of the First Peoples' Hospital of Jining approved this study (Approval number: JY2016035).

### Data Availability

The raw data are available as Supplemental Files.

### Supplemental Information

Supplemental information for this article can be found online at http://dx.doi.org/10.7717/peerj.9286#supplemental-information.

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
