# Peer review of "Association of FKBP5 polymorphisms with patient susceptibility to coronary artery disease comorbid with depression"

_PeerJ, doi:10.7717/peerj.9286_

## Round 0.1 · original submission · Major Revisions

Dear authours:
Your ms has now been reviewed by two leading experts.

Both found merit in the paper, both require major changes that will require rewriting the introduction, adding some analyses, revising the tables for greater clarity etc.

In addition, as editor, I ask you to carefully revise the language of the paper. Throughout the ms, starting with the first sentence of the abstract, a native English speaker should line-edit the manuscript.
I hope you undertake the revisions and resubmit.

When you do please take care to write the rebuttal letter in a detailed organized manner so that it is easy to re-review. This will make the processing of the ms faster, and make it much easier to re-review.

Reviewer 1 ·

Basic reporting

This study investigates the potential association between 7 SNPs within the FKBP5 gene and depression in a population of 123 patients with coronary artery disease (CAD) and comorbid depression, 147 CAD patients with no depression and 113 healthy controls. A significant association was found for rs9470079- GA genotype conferred decreased risk for CAD. Further associations were reported for two additional SNPs with sex effect on the association (significant for females but not for males).

Experimental design

The strengths of the study are the reasonable sample size and the recruitment of healthy controls, the standard clinical assessment and measures for depression, and the extensive study of a single gene- looking into seven different SNPs.

Validity of the findings

The limitations of the study are the narrow focus on a single gene:
1. To my knowledge candidate gene approach is not scientifically popular anymore. Instead the GWAS approach is more standard. I understand that the sample size is underpowered for conducting GWAS. However, the authors should refer to results of GWAS studies of depression and possibly also of CAD in the paper.
2. Are any of the SNPs with any known functional significance on the protein or are they all synonymous change with no effect on the protein? If the latter is true what is the significance of the findings?

Additional comments

no comment

Reviewer 2 ·

Basic reporting

The authors designed a case-control study, that included seven polymorphisms of FKBP51 gene in susceptibility to comorbid CAD and depression. I suggest modifying tables 3 to 6 as they are saturated with results and it is confusing to understand them. However, the information presented from the associations is correct.

Experimental design

The experimental design is adequate considering the objective of the study.

Validity of the findings

No comment.

Additional comments

It would be interesting to perform the linkage disequilibrium analysis and include possible haplotypes that are associated with the disease.

---

## Round 0.2 · accepted · Accept

Dear authors:

I am happy that your manuscript has now been re-reviewed and accepted.

Thank you for undertaking the revisions and doing such a good job.
Ada

Reviewer 1 ·

Basic reporting

The authors have revised the manuscript to meet my comments. I have no further comments

Experimental design

.

Validity of the findings

.

Additional comments

.

Reviewer 2 ·

Basic reporting

The revised manuscript is now more interesting.

I don't have more questions.

Experimental design

The experimental design is adequate with the hypothesis of the paper.

Validity of the findings

The authors are aware of the limitations of the study; however, they are working to improve future manuscripts